# Influence of Treated Distillate Aromatic Extract (TDAE) Content and Addition Time on Rubber-Filler Interactions in Silica Filled SBR/BR Blends

**DOI:** 10.3390/polym13050698

**Published:** 2021-02-25

**Authors:** Selin Sökmen, Katja Oßwald, Katrin Reincke, Sybill Ilisch

**Affiliations:** 1Polymer Service GmbH Merseburg, Eberhard-Leibnitz-Straße 2, 06217 Merseburg, Germany; katja.osswald@psm-merseburg.de (K.O.); katrin.reincke@psm-merseburg.de (K.R.); 2Trinseo Deutschland GmbH, Street E 17, 06258 Schkopau, Germany; SIlisch@trinseo.com

**Keywords:** silica, SBR, BR, rubber–filler interactions, viscoelastic properties, mechanical properties, Payne effect

## Abstract

High compatibility and good rubber–filler interactions are required in order to obtain high quality products. Rubber–filler and filler–filler interactions can be influenced by various material factors, such as the presence of processing aids. Although different processing aids, especially the plasticizers, and their effects on compatibility have been investigated in the literature, their influence on rubber–filler interactions in highly active filler reinforced mixtures is not explicit and has not been investigated in depth. For this purpose, the influence of treated distillate aromatic extract (TDAE) oil content and its addition time on interactions between silica and rubber chains were investigated in this study. Rubber–filler and filler–filler interactions of uncured and cured silica-filled SBR/BR blends were characterized by using rubber layer *L* concept and dynamic mechanical analysis, whereas mechanical properties were studied by tensile test and Shore A hardness. Five parts per hundred rubber (phr) TDAE addition at 0, 1.5, and 3 min of mixing were characterized to investigate the influence of TDAE addition time on rubber–filler interactions. It was observed that addition time of TDAE can influence the development of bounded rubber structure and the interfacial interactions, especially at short time of mixing, less than 5 min. Oil addition with silica at 1.5 min of mixing resulted in fast rubber layer development and a small reduction in storage shear modulus of uncured blends. The influence of oil content on rubber–filler and filler–filler interactions were investigated for the binary blends without oil, with 5 and 20 phr TDAE content. The addition of 5 phr oil resulted in a slight increase in rubber layer and 0.05 MPa reduction in Payne effect of uncured blends. The storage tensile modulus of vulcanizates at small strains decreased from 13.97 to 8.28 MPa after oil addition. Twenty parts per hundred rubber (phr) oil addition to binary blends caused rubber layer *L* to decrease from 0.45 to 0.42. The storage tensile modulus of the vulcanizates and its reduction with higher amplitudes were incontrovertibly high among the vulcanizates with lower oil content, which were 13.57 and 4.49 MPa, respectively. When any consequential change in mechanical properties of styrene–butadiene rubber (SBR)/butadiene rubber (BR) blends could not be observed at different TDAE addition time, increasing amount of oil in blends enhanced elongation at break, and decreased Shore A hardness and tensile strength.

## 1. Introduction

Silica-filled rubber blend technology is of interest to the elastomer industries, especially to the tire manufacturers due to its potential usage for tire tread formulations [1,2]. Chemical structure and microstructure of polymers, filler types, and filler characteristics mainly determine the type and strength of polymer–filler and filler–filler interactions [3]. Silica is compatible to some degree with polar rubbers due to its highly polar structure, but it is not naturally compatible with non-polar rubbers such as styrene–butadiene rubber (SBR), butadiene rubber (BR), and natural rubber (NR). The interfacial interactions between polar silica and non-polar hydrocarbon polymers are weak, yet the compatibility can be increased by the silica surface modification or the polymer chain functionalization. Coupling agents are used for chemical modification of silica surface by reducing the polarity difference between rubber and filler [4,5,6,7,8,9]. Previous studies have claimed that silane enhances interfacial interactions and improves dispersion of silica in rubber matrix [6,7,8,9,10]. The functionalization of polymer chains is likewise used to diminish polarity difference. Based on the results of studies with chain-end-modified and backbone modified polymers, modification of polymer chains can lead to an improvement of dynamic characteristics and mechanical properties [11,12].

Investigations of polymer–filler interactions are usually carried out by using different concepts, such as bound rubber and wetting concept. Bound rubber is defined as the structure formed by the attachment of polymer chains to the silica surface and this concept is commonly used to investigate polymer–filler interactions in filled rubber systems [13,14,15,16,17]. In the previous studies, bound rubber mechanisms and the factors affecting the bound rubber formation were extensively investigated [18,19,20,21,22]. Increase in bound rubber content with the presence of silane, increasing filler content and mixing time was observed [21,22], when it decreased at high extraction temperatures [20]. In binary and ternary rubber blend systems, the wetting concept was widely used to identify filler localization and rubber–filler interactions [23,24,25,26]. The bounded rubber principle lies under the wetting concept as well; however, the calculations are carried out differently from the bound rubber concept. Rubber layer *L* is calculated as a part of wetting concept and it provides information about the bounded rubber layer on the filler surface due to physical and chemical interactions. In the previous studies, it was claimed that rubber layer development continues until rubber infiltrates filler completely, where rubber–filler interactions reach the maximum amount. The rubber layer *L* remains constant at this maximum amount which is the plateau value ($L_{P}$), for a certain mixing time [24]. In addition to the concepts with bounded rubber structure, dynamic mechanical analysis was used to analyze rubber–filler interactions in rubber blends. Loss modulus (G″) was used to determine filler localization in binary blends, in which the increase of the peak height of the loss modulus in a corresponding phase was stated as an indicator of the filler localization in that phase [27,28]. In addition, different models were studied to investigate the contributions of filler network and hydrodynamic effects to storage modulus in reinforced systems and rubber–filler interaction as a contributor to dynamic modulus, was indirectly examined in these researches [29,30].

Processing oils are used in blends as plasticizers to improve flow properties and processability. Processing oils provide lubrication between polymer chains and do not cause any chemical changes. Nowadays, processing oils with high aromatic content are non-utilizable due to the environmental concerns. They are mainly replaced by low aromatic content oils, such as treated distillate aromatic extract oil (TDAE), and bio-oils. In the previous studies, the compatibility of processing oils in different polymer systems were studied and their influence on dynamic and mechanical properties were the main focus of these studies considering the commercial applications [31,32,33,34]. Although a few studies on TDAE oil in SBR blend systems are present in the literature, the localization and material characterizations were focused on the unfilled blend systems [35,36]. The plasticizers are commonly used with highly reinforced polymer systems in commercial applications, yet there are not enough studies in the literature for these kind of systems. The plasticizer influence on rubber–filler and filler-filler interactions in reinforced blends is not explicit and detailed investigations are required in this topic. Therefore, the focus of this work is on the TDAE influence in silica-filled SBR/BR blends.

In the present work, the influence of TDAE content and its addition time on rubber–filler interactions were studied in silica-filled SBR/BR binary blends. Rubber–silica interactions of the blends were examined by rubber layer *L* concept, whereas silica–silica interactions and dynamic mechanical properties were investigated by rubber process analyzer (RPA) and dynamic mechanical analysis (DMA). Mechanical analysis and microscopy analysis were carried out as supportive methods to study rubber–filler and filler–filler interactions and to estimate a possible correlation with rubber layer of silica-filled blends.

## 2. Experimental

### 2.1. Materials and Compounding

Solution styrene–butadiene rubber (S-SBR), SPRINTAN^TM^ SLR 4602 (Trinseo GmbH Schkopau, Germany) which is functionalized for improved polymer-filler interaction with carbon black and silica, and high cis-butadiene rubber (BR), BUNA^TM^ CIS 132 (Trinseo GmbH Schkopau, Germany) were used for compounding of binary blends. The compound formulation is given in Table 1. Rubber compounds were prepared by a laboratory internal mixer Haake 300p (Corp. Thermo Fisher GmbH, Dreieich, Germany). Rotor speed and fill factor were kept constant at 50 rpm and 60%, respectively. Mixing was carried out in two steps. In the first stage of mixing, SBR and BR were mixed with all ingredients, except the curatives. Silica ULTRASIL 7000 GR with CTAB surface area of 160 m^2^/g, (Evonik GmbH, Essen, Germany) was divided into two parts and fed into the mixing chamber. In situ silanization carried out by silane coupling agent, Si75 (Evonik GmbH, Essen, Germany) in the temperature range from 130 to 150 °C. Starting mixing temperature of the first stage of mixing was set to 130 °C considering to the optimized temperature of silanization reaction. The dump temperatures of the blends without TDAE, and 5 and 20 phr TDAE were 146, 144, and 140 °C, respectively. Processing oil, which includes reduced amount of polycyclic aromatic carbons up to 2.8% weight percent [37], treated distillate aromatic extract (TDAE) of the type Vivatec500 (Hansen & Rosenthal KG, Hamburg, Germany) was added to the system in this step. In the second stage of mixing, curing additives were added at 50 °C starting temperature. Sulfur (Carl Roth GmbH + Co. KG, Karlsruhe, Germany), diphenyl guanidine (DPG) (Carl Roth GmbH + Co. KG, Karlsruhe, Germany), *tert*butyl-2-benzothiazolsulfenamide (TBBS) (Carl Roth GmbH + Co. KG, Karlsruhe, Germany), and *n*-cyclohexylbenzothiazole-2-sulphenamide (CBS) (Carl Roth GmbH + Co. KG, Karlsruhe, Germany) were mixed with silica-filled SBR/BR blends in the mixing chamber for 5 min.

### 2.2. Sample Preparation and Characterization

Investigation of rubber–filler interactions of unvulcanized silica-filled SBR/BR blends was carried out by using rubber layer *L* concept. Approximately 1 g of samples started to be taken out of the chamber after completing the addition of all compounds, except curatives. Rubber samples were collected at different mixing times to examine the development of rubber layer. 0.2 g uncured silica-filled samples were extracted in 100 mL toluene solvent for 1 week. The solvent was changed once after 3 days. After the extraction, rubber–filler gel remained which is the undissolvable rubber structure consisting of rubber chains bonded on the silica surface. The bonded rubber layer is assumed to be formed by the filler network mediated or connected through the polymer chains and it cannot be extracted by the solvent extraction. The gel was dried in an air-circulated oven at 70 °C for 2 h and the samples were weighed. The numerical calculation of rubber layer *L* was carried out by using the Equation (1) [23]:(1)$$L=\;\frac{m_{2}-\;m_{1}\;c_{S}}{m_{2}}$$

In the Equation (1), *L* represents rubber layer *L*, where $m_{1}$ is the mass of the SBR/BR blend sample before toluene extraction and $m_{2}$ is the mass of the dried rubber–filler gel after the extraction. $c_{S}$ represents the weight fraction of silica in rubber–silica compounds. The maximum calculated rubber layer *L* which stays constant at this plateau value and known as $L_{P}$, is used as an indicator of maximum achievable rubber–filler interaction of the rubber blend.

Viscoelastic properties of unvulcanized blends were analyzed by rubber process analyzer (RPA) (Scarabaeus Mess- und Produktionstechnik GmbH, Wetzlar, Germany). Storage shear modulus was determined by amplitude sweeps between 0.1% and 100%, with 10 Hz frequency, and at 80 °C test temperature.

Light microscopy analyses were carried out by an optical microscope (Leica Microsystems GmbH, Wetzlar, Germany). Uncured silica-filled SBR/BR blends were analyzed to have a general understanding on macro scale dispersion and distribution of silica in rubber matrices.

Vulcanization characteristics of the silica-filled SBR/BR blends were characterized at 160 °C by using a vulcameter Elastograph (Göttfert Werkstoff Prüfmaschinen GmbH, Buchen, Germany) in accordance with DIN 5329-2 [38] and vulcanization time $t_{90}$ was determined for each compound. The compounds were vulcanized at 100 bar and at 160 °C as long as their $t_{90}$ time by using a laboratory press PM 20/200 (Fa. Campana Ing. Benedetto, Italy).

Crosslink density of SBR/BR vulcanizates were determined by swelling test. One gram of rubber samples were immersed in toluene solvent for 72 h. The swollen samples were taken out of the solvent and dried at room temperature for 4 days. The initial and final weights were measured, and the value of swelling degree was calculated by the Equation (2) [39,40]:(2)$$Q_{w}=\;\frac{m_{sp}-m_{s}}{m_{s}^{*}}$$

$Q_{w}$ is the equilibrium swelling, $m_{sp}$ is the mass of swelling sample, and $m_{s}$ is the mass of dried sample after swelling. $m_{s}^{*}$ is the mass of dried sample after swelling, corrected for the filled sample by Equation (3): (3)$$m_{s}^{*}=\;m_{o}-\left(\frac{m_{n}}{m_{c}}\right)$$
where $m_{o}$ is the initial mass of sample, *m**_n_* is the mass of mineral substances contained in the blend, and $m_{c}$ is the mass of all blend components. 

$V_{r}$ which is the volume fraction of the polymer in the vulcanizate swollen was determined by using the Equation (4):(4)$$V_{r}=\;\frac{1}{1+Q_{w}\;\left(\frac{d_{k}}{d_{r}}\right)}$$

In the Equation (4), $d_{k}$ is the density of rubber and $d_{r}$ is the density of of solvent. 

The crosslinking density according to the Flory–Rehner equation, given by the Equation (5), was determined [41].
(5)$$v=\;-\frac{\ln\left(1-V_{r}\right)+V_{r}+\;\chi \;V_{r}^{2}}{V_{0}\;\left(V_{r}^{\frac{1}{3}}-\;\frac{2V_{r}}{f}\right)}$$

Crosslinking density of vulcanizates (v) was calculated with the unit of mol/cm^3^, where χ is Flory–Huggings interaction parameter which was taken 0.378 for SBR in toluene and $V_{0}$ is the molar volume of solvent, 106.9 cm^3^/mol for toluene. f is the functionality of crosslinks, which was taken 4 by assuming the formation of tetra-functional crosslinks [42].

Dynamic mechanical analysis (DMA) of SBR/BR vulcanizates was performed in tension mode by DMA GABO EXPLEXOR (Netzsch Gerätebau GmbH, Selb, Germany) with a force head of 150 N according to DIN 53513 [43]. The specimens were tested by strain sweeps between 0.01% and 11%, with 10 Hz frequency and at 60 °C test temperature. Evaluation of the tensile properties of the vulcanizates was performed by following DIN 53504 standard by using S2 type specimens and the test speed of 200 mm/min [44]. Five specimens from each blend were tested for tensile test, and the average values were taken as the indication of tensile properties. Shore A hardness of silica-filled SBR/BR vulcanizates was determined according to DIN ISO 7619-1 [45]. Ten measurements from each blend were carried out and the average values were used as Shore A hardness.

## 3. Result and Discussion

### 3.1. Influence of TDAE Addition Time on Mixing Behavior and Interfacial Interactions

Mixing regime can result in different bound rubber development of rubber blends [46]. In Figure 1, mixing behavior of silica-filled SBR/BR blends with 5 phr TDAE in, is shown for different plasticizer addition time. Five phr oil content was chosen considering the easy handling of low plasticizer contents in mixing. Each blend was mixed for 10 min, after rubber fed to the mixing chamber which is shown as shaded areas on the graph. To investigate the plasticizer addition at 0 min, TDAE was fed to the chamber together with SBR and BR for 2 min to avoid sudden torque decreases. It is visible that mixing torque is, to a minor extent, lower compared to other blends at the first minutes of mixing and after feeding of silica at the second min. At the end of 10 min of mixing, torque is observed at the same level with other blends and dump temperature is 143 °C. For the investigation of plasticizer addition at 1.5 min, TDAE was fed to system after a short pre-mixing with silica and silica addition caused a little higher torque increase than other blends in the mixing. Dump torque and temperatures were 10 Nm and 144 °C, respectively. Addition of plasticizer at 3 min was carried out by a syringe, similar to 1.5 min addition. After the oil addition at 3 min, torque and temperature decreased slightly. Dump torque and temperatures were observed a little higher than other blends, which are 12 Nm and 145 °C, respectively.

The rubber layer *L* in dependence on mixing time was determined for silica-filled SBR/BR binary blends with different TDAE addition time, shown in Figure 2. The main influence of plasticizer addition time was observed on rubber layer *L* development at earlier stages of mixing. TDAE addition at 0 min showed the slowest rubber layer *L* development compared to 1.5 and 3 min of oil addition time. The reason could be the low viscosity by the addition of plasticizer at the beginning of mixing, that can influence the filler dispersion [47] and can cause poor dispersion at short time of mixing. Retarded reduction of agglomerates because of the small shear forces during mixing may delay the completion of the rubber layer formation. The fastest rubber layer development was observed for TDAE addition at 1.5 min with silica. On the other hand, any significant difference could not be seen in the plateau values ($L_{P}$) of binary blends, which are around 0.4 for each. Since $L_{P}$ represents the maximum achievable rubber layer in the blend, it can be claimed that approximately 10 min of mixing is required in this mixer with the described mixing parameters to achieve maximum rubber–filler interactions for the SBR/BR blends without depending on the oil addition time.

Filler–filler interactions and flocculation behavior can be determined with the help of viscoelastic property investigations. The storage shear modulus (*G*′) at low strain amplitude is used as a measure for the filler–filler interactions and its decrease with increasing dynamic strain amplitudes is used to characterize the reinforcing mechanism of filler in rubber compounds [48]. *G*′ of uncured silica-filled SBR/BR binary blends with different TDAE addition time is shown in Figure 3. The influence of plasticizer addition time can be seen on *G*′ at low strain amplitudes. Although all uncured silica-filled blends show a similar trend, plasticizer addition at 1.5 min of mixing results in slightly lower *G*′ compared to others. When lower *G*′ at low strains is considered as a representation of less silica flocculation [49], it can be said that the binary blend with TDAE added at 1.5 min forms less filler–filler networking and less flocculation. According to these results combined with the rubber layer *L* analysis shown in Figure 2, it can be declared that TDAE addition together with silica at 1.5 min of mixing provided advantages to achieve less flocculation and more interfacial interactions between silica and SBR/BR matrices.

### 3.2. Influence of TDAE Content on Rubber–Filler and Filler–Filler Interactions

In Figure 4, the development of rubber layer *L* in silica-filled SBR/BR blends can be seen depending on TDAE concentration for different mixing times. The binary blend without plasticizer showed fast rubber layer *L* development at early stages of mixing and resulted in approximately same $L_{P}$ value with 5 phr TDAE added blend. It is known that the development of rubber layer on the filler surface is not only depending on the rubber–filler compatibility, but also mixing conditions [50]. High filler concentration and the absence of processing oil result in pronounced increase in shear forces during mixing, which may improve silica dispersion in rubber matrix by breaking down silica agglomerates [51]. SBR/BR blend with 5 phr TDAE showed the fastest rubber layer development. Low oil content can improve the dispersion of silica in matrix by helping to lubricate rubber chains [35,36] and in principle, improvement in dispersion during mixing can cause fast development of the rubber layer and may result in more rubber–filler interactions. On the other hand, the reason of the slow development with the addition of 20 phr TDAE can be the plasticization effect of oil that leads to the decreased bulk viscosity [31]. It might delay the filler dispersion and decelerate rubber layer development due to the small internal shear forces during mixing with the presence of high oil content. The $L_{P}$ value of the blend with 20 phr TDAE was observed slightly lower compared to other blends after 10 min of mixing. It can be claimed that the presence of high content of plasticizer does not only delay the rubber layer formation, but also it reduces the bonded rubber layer on silica surface. A possible explanation for this could be the penetration of TDAE into interstices between rubber and silica [32] that can partially block the interactions between polymer chains and silica. According to the results of TDAE content influence in silica-filled SBR/BR blends, it can be claimed that high plasticizer contents can require longer mixing time to achieve maximum rubber–filler interactions in the system and result in reduction in bonded rubber layer on fillers.

Microscopy analysis of macro-structures and micro-structures of fillers can be utilized to examine rubber–filler interactions, beside filler–filler interactions. In high level of silica loadings, qualitative analysis of micro-dispersion and micro-structures are difficult because of the three-dimensional superposition of the aggregates inside agglomerates [52]. In addition, the presence of oils and various components typically used in tire technology, make the analysis more complex. Macro-scale investigations of uncured SBR/BR blends were carried out by optical light microscopy to observe silica dispersion and distribution, shown in Figure 5. Silica aggregates and agglomerates with different sizes can be seen as dark regions in SBR/BR matrix, in Figure 5a. The spherical structures up to 40 µm diameter which are marked with arrows, can be silica agglomerates and indicates poor dispersion. In principle, the big silica agglomerates are destroyed under high shear forces during mixing and the probability of presence of flocculated silica particles after long mixing time like 10 min is low, however the silica can re-agglomerate after mixing. Considering to the high tendency of re-agglomeration of silica after mixing [9], analyses can be influenced by the re-agglomeration during storage time. The presence of 5 phr TDAE in the binary blend caused a noticeable increase of the dark regions and the amount of spherical structures in the images, seen in Figure 5b. The enhancement of dark regions is observed much higher for 20 phr plasticizer added blend, beside blurred dark regions shown by arrows in Figure 5c. It is hard to distinguish the regions belonging to silica agglomerates and TDAE oil droplets in light microscopy images, since both silica agglomerates and TDAE droplets appear dark under light microscopy. The presence of high amount of oil can shield the silica aggregates as well and it is not possible to make a statement on filler flocculation. These results demonstrate the need for other characterization methods for the investigation of filler–filler interactions. For this purpose, Payne effect was investigated for the unvulcanized binary blends.

Rubber trapped in filler aggregates called occluded rubber, contributes to the filler network behaving like filler. The breakdown of the filler network with increasing amplitude may release the trapped rubber for energy dissipation and the released rubber can take part in deformation again. It results a non-linear decrease in the shear modulus and this decrease is named Payne effect [53,54]. It is often measured by the difference between the low amplitude modulus and the high amplitude modulus [55]. In this work, Payne effect of uncured blends was calculated by the difference between storage shear modulus at 0.1% and 100% strain amplitudes (Δ*G*′*_0.1%–100%_*). Storage modulus can also be utilized to understand the occluded rubber and rubber–filler interactions. At high strains, filler-filler network breaks and the storage modulus becomes strain independent. The strain independent modulus consists of different contributions such as hydrodynamic effect and in-rubber structure, which is emphasized as a direct measure for the occluded rubber [56]. For this purpose, the storage modulus at 100% strain (*G*′*_100%_*) in which the filler–filler network was completely broken, was examined additionally in this study. In Figure 6, Δ*G*′*_0.1%–100%_*, *G*′*_100%_* and the rubber layer $L_{P}$, dependence on TDAE content were shown for silica-filled SBR/BR blends. Reduction in Payne effect and *G*′*_100%_* were observed with increasing TDAE content in blends. Decrease in Payne effect and slight increase in $L_{P}$ may be an indication of better silica dispersion [57,58] and less flocculation. However, it is important to emphasize that occluded rubber is not considered for the comments on rubber–filler interactions calculated by rubber layer *L*. In addition, the presence of oil influences the behavior of *G*′ and all contributions should be calculated to understand the amount of occluded rubber by *G*′*_100%_.* Highest reduction in Payne effect and *G*′*_100%_*, beside $L_{P}$ were observed after 20 phr oil addition, which may not indicate better silica dispersion. In this case, the plasticization effect of oil which causes a reduction in storage modulus, can be considered as the reason of decreasing Payne effect. As high oil concentrations decrease the viscosity, the reduction of silica agglomerates into aggregates is not highly probable during mixing due to the less shear forces. In addition, good filler dispersion is not expected in less viscose matrix, considering to high agglomeration tendency of silica. Another possible explanation can be the location of TDAE between rubber chains and silica. For high oil concentrations, it was claimed that the penetration of oil into interstices between rubber chains and fillers can influence the interactions between mutual filler particles and cause weaker interactions [32]. In this case, reduced Payne effect can be observed. Parallel to this result, $L_{P}$ of 20 phr oil added blend was the lowest which indicates a reduction in rubber–filler interactions.

### 3.3. Investigation of Dynamic Mechanical and Mechanical Properties of SBR/BR Vulcanizates and Their Relation to Rubber Layer L

Material properties of rubber blends are highly influenced by the density of crosslinks within the vulcanizates. The crosslink densities of silica-filled SBR/BR vulcanizates were quantitatively analyzed with the swelling test and Flory–Rehner equation, shown in Figure 7. Vulcanizates with 5 phr TDAE content showed the highest crosslink densities independent of their addition time, and they are followed by 20 phr TDAE content. When TDAE addition at 0 and 1.5 min had the crosslinking densities around 3.5 × 10^−5^ mol/cm^3^, addition at 3 min resulted in a slightly higher value, 3.8 × 10^−5^ mol/cm^3^. The enhanced crosslink density with 5 phr oil addition indicates dense internal structure in the vulcanizates, in which silanized silica and polymer chains bonded with sulfur more efficiently compared to the other blends. High density of crosslink may be an indication of high rubber–filler interactions [59], which fits to the previous rubber layer and viscoelasticity results of unvulcanized blends after 5 phr oil addition. 20 phr TDAE addition resulted in smaller crosslink density compared to 5 phr oil content. It agrees that the high oil concentrations can block the interfacial interactions and formation of crosslinks. On the other hand, the crosslink density of the 20 phr TDAE added vulcanizate was higher than the vulcanizate without oil, which are 2.6 × 10^−5^ mol/cm^3^ and 1.3 × 10^−5^ mol/cm^3^, respectively. 

Dynamic mechanical properties were characterized to investigate the influence of TDAE content and addition time on SBR/BR vulcanizates. In Figure 8, storage tensile modulus (*E*′) and loss tensile modulus (*E″*) of silica-filled SBR/BR vulcanizates are shown for different TDAE contents and addition time. It can be seen in Figure 8a that the absence of TDAE resulted in the highest *E*′, 13.97 MPa, and was followed by 20 and 5 phr TDAE addition, 13.57 and 8.28 MPa, respectively. Reduction in storage modulus is expected with increasing oil contents due to their viscous nature and a certain softening effect, nevertheless 20 phr oil addition to vulcanizate caused a small reduction in *E*′ and more pronounced storage modulus decrease with increasing strains, 4.49 MPa. It is the indication of high silica flocculation in the vulcanizate, even though the unvulcanized blend showed smaller Payne effect under shear forces, as an indicator of less filler flocculation (see Figure 6). TDAE may indirectly affect the curing efficiency and can cause differences in behavior of SBR/BR vulcanizates compared to unvulcanized blends. The presence of high oil contents may fill the moieties of filler surface and cause the penetration into interstices between filler and rubber. In this case, TDAE oil can influence the silanization efficiency [60]. Further, the silanization efficiency which affects the amount of free sulphur bonds, influences the curing efficiency [61]. Five parts per hundred rubber (phr) oil added vulcanizates exhibited low *E*′ in Figure 8a, and TDAE addition at 1.5 min of mixing resulted in the least *E*′, similar to the previous results. An inversely proportional relationship between *E*′ and $L_{P}$ which is given in Table 2, was observed for all 5 phr TDAE added blends. It indicates that at same amount of oil content, less filler–filler networking in blends were resulted in higher $L_{P}$ and more rubber–filler interactions. The possibility of improvement in filler dispersion can be supported by both Payne effect and $L_{P}$ values. Decrease in Payne effect and slight increase in $L_{P}$ may be an indication of less filler–filler networking and more rubber–filler interactions. *E″* of silica-filled SBR/BR vulcanizates were shown depending on oil content and addition time in Figure 8b. Twenty parts per hundred rubber (phr) oil added vulcanizate exhibited the highest energy dissipation due to its high oil content. SBR/BR vulcanizate without oil showed higher energy dissipation than 5 phr oil added vulcanizates. It can be simply explained with the higher amount of immobilized rubber with higher molecular friction, and thus higher energy dissipation [62].

The loss factor tan *δ* at 60 °C was calculated as the ratio of loss modulus (*E″*) to storage modulus (*E*′). It is known that lower values of tan *δ* indicate lower hysteresis, which results in less energy dissipation [53]. Figure 9 shows the tan *δ* at 60 °C values of the silica-filled SBR/BR vulcanizates depending on TDAE content and addition time. It can be seen that the SBR/BR blends without TDAE and with 5 phr TDAE added at 0 and 3 min of mixing, show quite close hysteresis and the value is lower than the blend with 20 phr oil content. The highest hysteresis belongs to 20 phr TDAE content due the contribution of its high *E*″ to tan *δ*, shown in Figure 8b. In addition, the interfacial slippage between rubber and filler, which is a source of energy dissipation [63] may be another reason. With the presence of high oil content in blends, the slippage of entanglements can be more and result in higher energy dissipation. On the other hand, the presence of 5 phr TDAE which was added at 1.5′ resulted in the lowest tan *δ* at 60 °C. This decrease in the hysteresis may be attributed to a less developed filler–filler network [53].

Good compatibility and homogeneous dispersion and distribution of filler in rubber blends affect the mechanical properties of materials [33,34,35,36]. The results of the mechanical property analysis of silica-filled SBR/BR blends with different TDAE content and addition time are presented in Table 2. It can be seen that increasing oil concentration caused an increase in elongation at break ($\varepsilon_{R}$) which indicates an improvement in elasticity. This behavior can be explained simply by plasticization effect of oil and the increasing flexibility of polymer chains [36]. Tensile stress of silica-filled SBR/BR binary blends with different oil contents showed a directly proportional relationship with $L_{P}$. Tensile strength ($\sigma_{M}$), tensile stress at 100% elongation ($\sigma_{100\%}$) and 200% elongation ($\sigma_{200\%})$ did not show any consequential change after 5 phr oil addition to the blend; however, it decreased with 20 phr oil content. Considering low $L_{P}$ values of 20 phr oil content, it can be claimed that presence of less bounded rubber structure and less rubber–filler interactions resulted in low tensile strength. Similar tensile behavior of high TDAE content was observed in the previous studies [32] and it can be explained by the dilution of the contact points between polymer chains with the presence of oil, which causes less resistance of filler aggregates and materials to the deformation. Shore A hardness showed a decreasing trend with increasing oil content. Due to the mechanical property analysis, TDAE addition time did not cause a remarkable difference in mechanical properties of materials and any structural relation to the mechanical properties could not be stated for 5 phr oil added blends.

## 4. Practical Applications and Future Research Perspectives

The silica-filled SBR/BR blends are mainly used for treads of passenger car tires [2]. The replacement of the carbon black by silica provides the improvement in rolling resistance and wet grip; however, the biggest challenge of these systems are the flocculation of silica and the compatibility of the fillers with polymers [1]. As in this study, high filler concentrations are used in commercial applications and modified SBR types, coupling agents and processing aids are used to overcome the problem of low compatibility. In large scale productions, high concentrations of plasticizers are used to reduce the viscosity and the energy cost, which complicate the system more. Complexity in tire formulations due to the presence of high amount of filler and processing aids, cause some difficulties and limitations in characterization techniques. Development of new qualitative and quantitative characterization methods for these systems is still open to research. In addition, development of ecofriendly plasticizer alternatives and their optimum usage to achieve a better compatibility and energy and cost effectiveness can be another research direction.

## 5. Conclusions

The influence of TDAE oil content and its addition time on rubber–filler interactions in silica-filled SBR/BR blends were investigated in this study. It was found that the addition time of TDAE influences the development of bounded rubber structure and the interfacial interactions at short time of mixing. Presence of 5 phr oil, especially its addition together with silica at 1.5 min of mixing, provided advantages to achieve less flocculation and more interactions between silica and SBR/BR matrices. Increase of oil content resulted in delayed rubber layer formation and reduction in its maximum value which indicates less rubber–filler interactions. Likewise, filler networking was observed stronger due to the dynamic mechanical analysis of binary vulcanizates. The presence of oil and its increasing concentration in blends enhanced elongation at break, while Shore A hardness and tensile strength decreased.

## Figures and Tables

**Figure 1 polymers-13-00698-f001:**
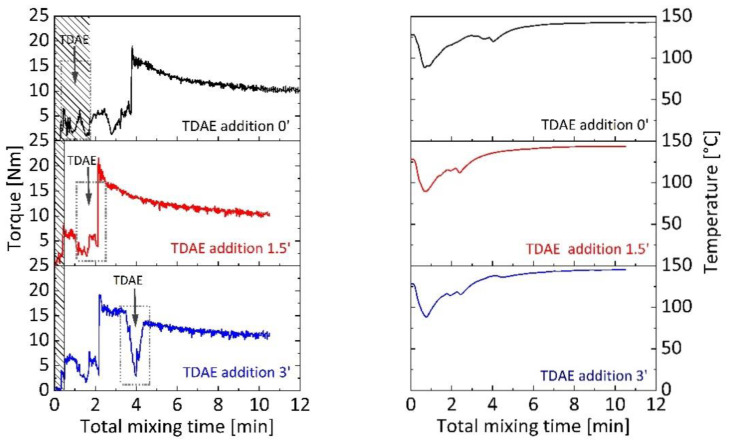
Mixing torque and temperature curves versus total mixing time for silica-filled styrene–butadiene rubber (SBR)/butadiene rubber (BR) binary blends with 5 phr Treated Distillate Aromatic Extract (TDAE) content at different addition times.

**Figure 2 polymers-13-00698-f002:**
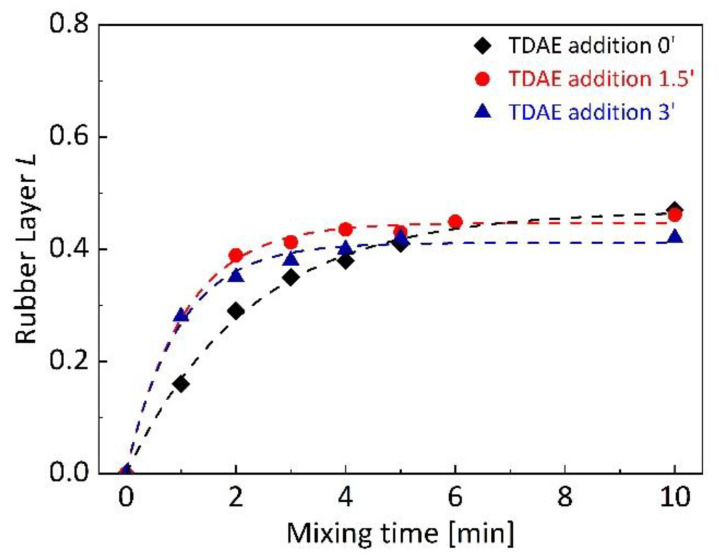
Dependence of rubber layer *L* development during mixing on TDAE addition time for silica-filled SBR/BR binary blends.

**Figure 3 polymers-13-00698-f003:**
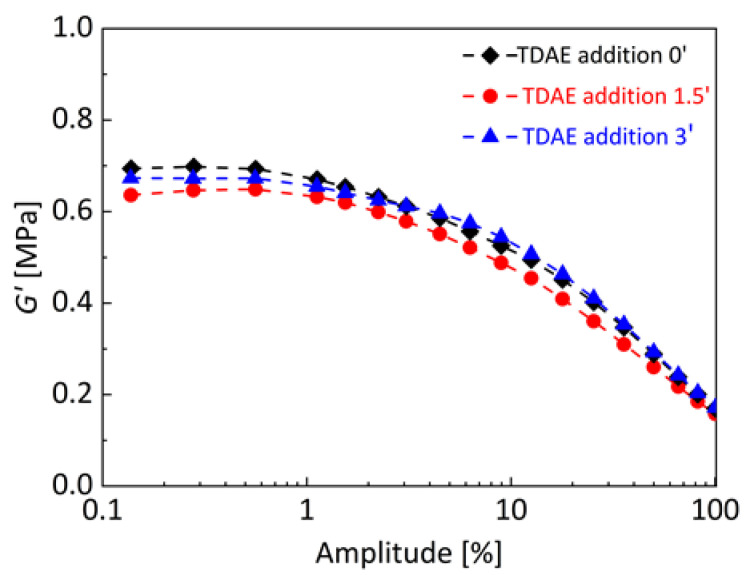
Storage shear modulus (*G*′) of uncured silica-filled SBR/BR binary blends at different TDAE addition time. The analyses were carried out with 10 Hz frequency and at 80 °C.

**Figure 4 polymers-13-00698-f004:**
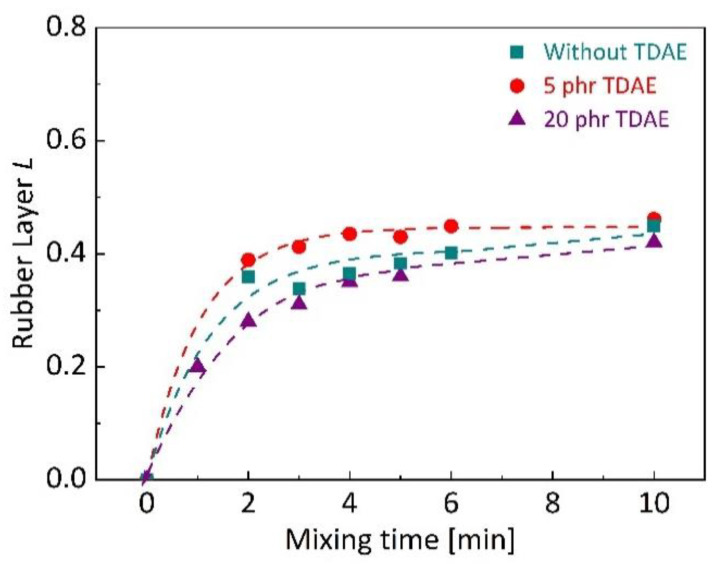
Dependence of rubber layer *L* development during mixing on TDAE content in silica-filled SBR/BR binary blends.

**Figure 5 polymers-13-00698-f005:**
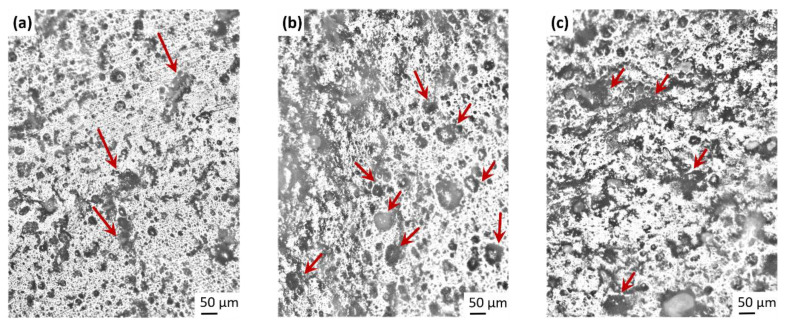
Light microscopy images of uncured blends after 10 min of mixing. Analyses at 100× overall magnification for silica-filled SBR/BR blends without (**a**); with 5 phr (**b**); and with 20 phr (**c**) TDAE content.

**Figure 6 polymers-13-00698-f006:**
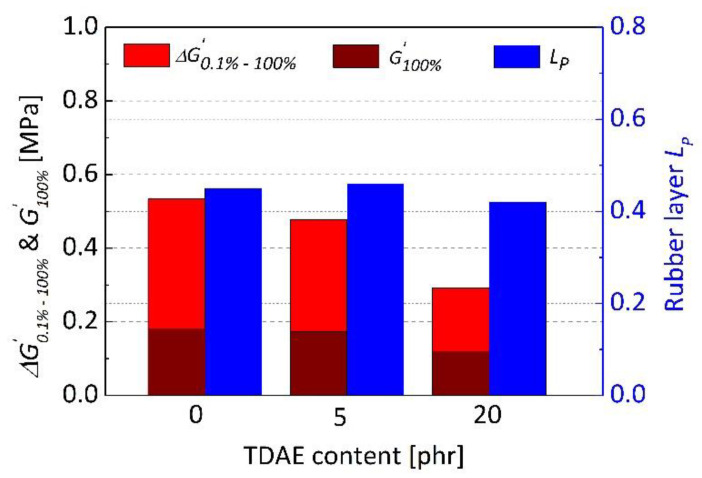
Difference between storage shear modulus at 0.1% and 100% strain amplitudes (Δ*G*′*_0.1%–100%_*), the storage shear modulus at 100% strain amplitude (*G*′*_100%_*) and the plateau value of rubber layer $\left(L_{P}\right)\;$relation of uncured silica-filled SBR/BR blends depending on TDAE content.

**Figure 7 polymers-13-00698-f007:**
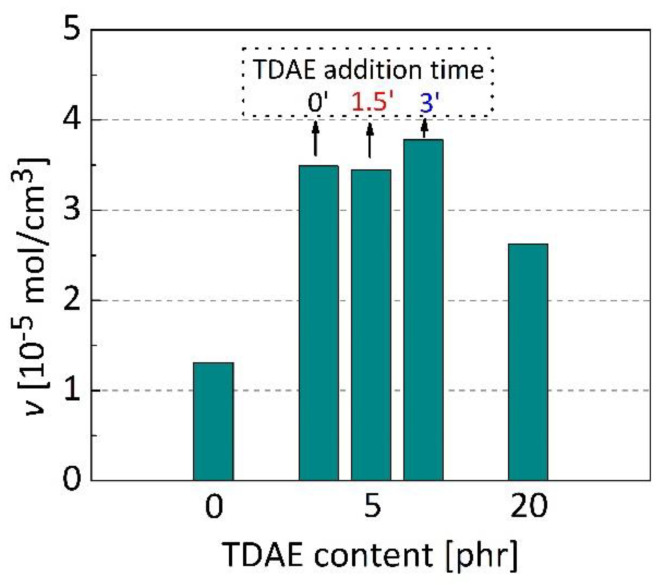
Crosslink density (v) of silica-filled SBR/BR vulcanizates with varying TDAE content and addition time, as calculated using the Flory–Rehner equation.

**Figure 8 polymers-13-00698-f008:**
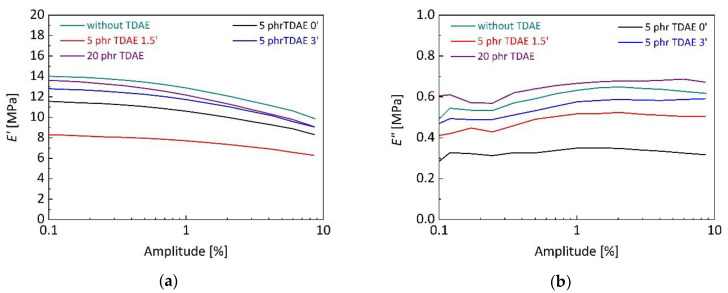
Storage tensile modulus (*E*′) (**a**); and loss tensile modulus (*E″*) (**b**) of silica-filled SBR/BR vulcanizates depending on TDAE content and addition time. The analyses were carried out with 10 Hz frequency and at 60 °C.

**Figure 9 polymers-13-00698-f009:**
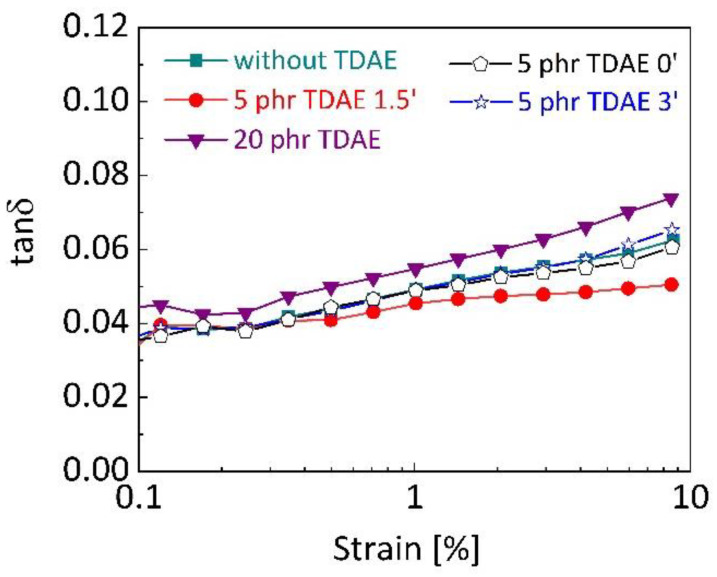
Mechanical loss factor (tan *δ*) of silica-filled SBR/BR vulcanizates depending on TDAE content and addition time. The analyses were carried out with 10 Hz frequency and at 60 °C.

**Table 1 polymers-13-00698-t001:** Compound formulation.

Components	Content, phr	Addition time, min
Stage I Mixing		
S-SBR	80	0
BR	20	0
Silica Ultrasil 7000	50	1 and 1.5
Silane Si75	4.2	1 and 1.5
TDAE	0/5/20	0/1.5/3
Zinc Oxide	0.5	0.5
Stearic Acid	3	0.5
End Time: 10 min		
Stage II Mixing		
Sulfur	2.5	0
DPG	1.5	0
TBBS	1.5	0
CBS	1.5	0
End time: 5 min		

**Table 2 polymers-13-00698-t002:** Rubber layer $L_{P}$ values of silica-filled SBR/BR binary blends and mechanical parameters of the binary vulcanizates.

	TDAE Content & Addition Time				
Properties	without TDAE	5 phr, 0 min	5 phr, 1.5 min	5 phr, 3 min	20 phr, 1.5 min
$L_{P}$	0.45	0.47	0.46	0.43	0.42
$\sigma_{100\%}$(MPa)	5.83 ± 0.11	5.22 ± 0.18	5.06 ± 0.28	5.21 ± 0.22	3.27 ± 0.05
$\sigma_{200\%}$(MPa)	13.6 ± 0.16	12.4 ± 0.12	11.4 ± 0.96	12.2 ± 0.13	7.55 ± 0.09
$\sigma_{M}$(MPa)	12.9 ± 2.57	12.4 ± 1.25	12.7 ± 1.14	13.2 ± 1.37	12.3 ± 0.96
$\varepsilon_{R}$(%)	191 ± 31.6	200 ± 14.0	218 ± 16.7	210 ± 17.5	281 ± 14.8
Hardness (Shore A)	73.7 ± 1.1	71.6 ± 0.6	70.6 ± 0.5	68.8 ± 0.4	65.2 ± 0.6

## Data Availability

Not applicable.

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
