# Peer review of "Influence of Treated Distillate Aromatic Extract (TDAE) Content and Addition Time on Rubber-Filler Interactions in Silica Filled SBR/BR Blends"

_polymers, 2021, doi:10.3390/polym13050698_

Round 1

Reviewer 1 Report

I read carefully the review article entitled ‘Influence of Treated Distillate Aromatic Extract (TDAE) Content and Addition Time on Rubber-Filler Interactions in Silica FilledSBR/BR Blends’. The concept of the manuscript fits and suitable to publish in Polymers Journal. This manuscript is generally well written and clearly presented however still need to address many comments and thus require substantial major revision.

  • Abstract section should rewrite looks very general and not informative. In abstract authors should mention the values of the results and importance of research work in one or two sentences. Give full form of abbreviations used in the manuscript somewhere.
  • Provide a nice graphical abstract representing the overview of the MS with key highlights.
  • In the introduction section, write the novelty of the work and the problem statement clearly. The authors fail to explain how this research work is suitable for Polymers give substantial discussion for the same.
  • Avoid cluster of references give details and detailed discussion about the novelty, significance of your research work and research gap relative to the literature.
  • In figure and table always give full form of abbreviation. In addition, for figure and table caption give all details.
  • Statistical analysis of the results should be provided in the materials and methods section. It's important for all experimental work Report these values in the results and discussion.
  • What are the chemical composition of treated distillate aromatic extract oil (TDAE) used in this study give details.
  • Have authors checked crosslink density of the produced blends? what are the observations? 
  • Surprisingly very less discussion of results with the previous results of literature need substantial discussion at revision stage. Use recent references from the year 2018-2020.
  • Write the practical applications and future research perspectives and challenges by adding a new section before conclusions.
  • What are the limitation to use this methodology for commercial application.
  • The conclusion of the study is not discussed with the specific output obtained from the study, it could be modified with precise outcomes with a take home message.
  • English and grammar mistakes are present. The author should check the manuscript by native English Speaker to improve the quality of the manuscript.

Author Response

Dear Reviewer 1,

Thank you very much for your time and valuable suggestions. You can find the new manuscript and the responses your comments.

Reviewer 2 Report

In the paper are presented a series of information regarding influence of treated distilled aromatic extract (TDAE) on rubber-filler interactions. From the analysis of the information presented in the article, I found the following:

- the paper presents a series of results that are not very important for the scientific community;

- the introductory part could be improved by adding information published in other research papers;

- the abbreviation CB must be specified. I think it's carbon black.

- the research methodology is rudimentary and very little detailed;

- table 1 shows compound formulation. From this table it is observed that ADHD can have values between 0 phr respectively 37.5 phr and addition time 0 / 1.5 / 3 min. According to the presented, a minimum of 15 samples should have been made. Unfortunately, this has not been done in research. For example, samples with 15, respectively 37.5 phr TDAE, are not analyzed. It must be detailed why this was not necessary !!!;

- it is necessary to present macroscopic and microscopic images of the samples used in the research;

- in the discussion part, the causes that determined the obtaining of those results in research must be better explained. A quantitative presentation of the results alone is not enough;

- in the final part of the conclusions, the future research directions and the possible practical applications of these materials must be presented.

Author Response

Dear Reviewer 2,

Thank you for your valuable time and comments. You can find the explanations and responses to your comments in the attached file.

Round 2

Reviewer 1 Report

Authors have substantially revised the manuscript according to the reviewers comments. The present form of the manuscript can be accepted.

Reviewer 2 Report

The authors revised their manuscript according to my suggestions. Thus the manuscript can be accepted for publication.